# Novel Au Nanoparticle-Modified ZnO Nanorod Arrays for Enhanced Photoluminescence-Based Optical Sensing of Oxygen

**DOI:** 10.3390/s23062886

**Published:** 2023-03-07

**Authors:** Baosheng Du, Meng Zhang, Jifei Ye, Diankai Wang, Jianhui Han, Tengfei Zhang

**Affiliations:** 1State Key Laboratory of Laser Propulsion and Application, Department of Aerospace Science and Technology, Space Engineering University, Beijing 101416, China; 2Institute of War Studies, Academy of Military Sciences, Beijing 100091, China

**Keywords:** Au nanoparticle, ZnO nanorod arrays, oxygen sensing

## Abstract

Novel optical gas-sensing materials for Au nanoparticle (NP)-modified ZnO nanorod (NR) arrays were fabricated using hydrothermal synthesis and magnetron sputtering on Si substrates. The optical performance of ZnO NR can be strongly modulated by the annealing temperature and Au sputtering time. With exposure to trace quantities of oxygen, the ultraviolet (UV) emission of the photoluminescence (PL) spectra of Au/ZnO samples at ~390 nm showed a large variation in intensity. Based on this mechanism, ZnO NR based oxygen gas sensing via PL spectra variation demonstrated a wide linear detection range of 10–100%, a high response value, and a 1% oxygen content sensitivity detection limit at 225 °C. This outstanding optical oxygen-sensing performance can be attributed to the large surface area to volume ratio, high crystal quality, and high UV emission efficiency of the Au NP-modified ZnO NR arrays. Density functional theory (DFT) simulation results confirmed that after the Au NPs modified the surface of the ZnO NR, the charge at the interface changed, and the structure of Au/ZnO had the lowest adsorption energy for oxygen molecules. These results suggest that Au NP-modified ZnO NR are promising for high-performance optical gas-sensing applications.

## 1. Introduction

Oxygen monitoring and quantification in gas mixtures and/or specific environments are of great significance in many fields, including life science and environmental quality, food storage, vacuum techniques, the automotive industry, the chemical industry, aerospace technology, and diving operations [1,2,3,4,5,6,7,8]. Real-time dynamic monitoring and quantification of oxygen over a wide range of concentrations is crucial for practical application. In recent decades, many researchers have conducted in-depth research to develop oxygen sensors [9,10,11,12,13,14]. Compared with the traditional electrical gas-sensing method, the photoluminescence (PL)-based optical gas-sensing method offers several distinct advantages, including good precision and accuracy, high sensitivity, good reversibility, non invasive measurement, no electrical contacts, ease of miniaturization, simple device fabrication, use in the presence of strong electromagnetic radiation, and remote operation [15,16,17,18]. Typical semiconductor oxide nanomaterials such as WO_3_ [19], TiO_2_ [20,21], ZnO [22,23], and SnO_2_ [24] have been successfully applied as PL-based gas-sensing materials due to their unique structure-related characteristics. In general, their sensing mechanism depends on the adsorption of gas releasing or capturing electrons from oxides, resulting in enhancement or quenching of the PL emission [19,25,26]. Although typical semiconductor oxides have low cost, controllable morphology, dopants, and good surface properties, the reported sensing response mechanism of oxides is limited in practical application [27,28,29,30]. Thus, it is of great scientific significance and application value to identify new PL-based gas-sensing oxides with significantly improved sensing capabilities.

ZnO is an n-type semiconductor with a large bandgap energy (3.37 eV), large exciton binding energy (0.06 eV), high chemical stability, excellent optical properties, non-toxicity, a low dielectric constant, and it is suitable for gas detection [31,32,33]. As one of the most promising optical gas-sensing materials, ZnO nanomaterials have controllable morphology, excellent near-band edge ultraviolet (UV) emission properties, high oxygen-adsorption capacity, and low-cost preparation methods [22,23]. Recently, extensive efforts have been devoted to the design and fabrication of ZnO-based optical gas sensors to increase their sensitivity [10,22,23]. It has been reported that ZnO nanomaterials can improve the UV-emission-based sensing performance of target gasses such as H_2_, O_2_, CO, H_2_S, and NO_2_ by controlling the morphology, crystal defects, doping, and surface modification [10,22,34,35,36,37,38,39]. However, many sensing performance issues remain unsatisfactorily resolved, such as low specific surface area, poor crystal quality, easy aggregation of sensing materials, deficient exposed active sites for target gases, insufficient UV emission efficiency, and surface inhomogeneity of sensing layers induced by preparation techniques. Thus, there is a need for a morphology- and structure-controllable, high-yield production method for ZnO nanomaterials with high UV emission intensities, and high UV emission-based optical gas sensors.

The electrical gas-sensing performance of ZnO and other oxides/hydroxides, such as CuO, SnO_2_, TiO_2_, and MoO_3_, can be considerably enhanced by Au nanoparticle (NP) modification [40,41,42,43,44]. Mechanistically, Au NPs are assumed to facilitate oxygen chemisorption on the oxide surface, enhancing gas-sensing properties [40,41,42]. Additionally, Au NPs have been shown to influence the structure, morphology, and particularly, the light emission properties of ZnO [45]. These findings have encouraged the investigation of Au NP-modified ZnO nanomaterials for potential optical gas sensor applications. More efforts are required to adequately engineer the role of Au NPs in enhancing gas-sensing performance, and to clarify the interrelation between the content, morphology, and structure of Au NPs, as well as the gas-sensing performance, which may offer guidance for the design of novel sensing materials.

The present work reports the growth of unique Au nanoparticle-modified ZnO NRs and provides insights into their likely formation mechanism. The preparation process (a) of Au/ZnO nanorod (NR) arrays on a Si substrate is shown in Figure 1. The PL properties of ZnO-based NR arrays with different annealing temperatures and Au sputtering times are investigated. Additionally, we explore the utility of these Au/ZnO NRs as PL-based sensors and their selectivity and sensitivity to oxygen detection. The results show that the intensity of the UV emission can be sensitively changed by oxygen adsorption, showing a high response, excellent selectivity, and high recoverability. In addition, the oxygen sensing mechanisms are analyzed in detail and further verified by density functional theory (DFT) simulation results.

## 2. Materials and Methods

### 2.1. Preparation of Au/ZnO NR Array Substrate

In this experiment, a ZnO seed layer was prepared on a Si substrate by magnetron sputtering. The silicon substrate was cleaned with acetone, oxygen, and deionized water. We started the mechanical pump and molecular pump to pump the vacuum to 3.0 × 10^−3^ Pa. At the same time, the heating system was turned on to heat the substrate to 400 °C. High-purity Ar gas at a flow rate of 40 sccm was introduced when the temperature stabilized. When the pressure in the chamber reached 5 Pa, the substrate was covered with a baffle, and the surface of the ZnO target was reduced with a radio frequency power of 90 W. High-purity O_2_ gas with a flow rate of 10 sccm was introduced. The substrate holder rotated to deposit the ZnO seed layer for 10 min when the baffle was opened. The distance between the targets and the substrate was 7 cm.

The water-bath growth process of ZnO NR arrays on the seed layer is described as follows. First, 50 mL of zinc nitrate and hexamethylenetetramine solutions with equal molar concentrations (0.1 M) were sealed in a reaction flask with a capacity of 100 mL. We placed them into a thermostatic water bath for 20 min at 90 °C. The Si substrate with the ZnO seed layer was placed into the mixed solution, and the temperature was maintained at 90 °C for 3 h in a sealed reaction flask. The prepared substrate was placed on a slide and washed with oxygen and ultrapure water. The samples were dehydrated and dried under moderate nitrogen flow.

Au NPs were deposited on the ZnO NR array substrate by radio frequency magnetron sputtering. The target was replaced with an Au target, and the ZnO NR array substrate was fixed on the substrate frame. The ZnO NR array substrates were sputtered with Au nanoparticles on a rotated substrate frame for 10 s, 30 s, 50 s, 70 s, and 90 s at a certain angle.

### 2.2. Material Characterization and Optical Gas-Sensing Measurements

The morphology and structure of the Au/ZnO samples were performed by scanning electron microscopy (SEM, JEOL 6700F) equipped with X-ray energy dispersive spectroscopy (EDS). The microstructural characteristics of Au/ZnO samples were investigated by transmission electron microscopy (TEM, TECNAI G2 F30 S-Twin). The crystalline phase of the Au/ZnO samples were identified using X-ray diffraction (XRD), using Cu Kα radiation at 40 kV in the range of 20–60°. The radiation source was a 150 W ozone free Xe lamp equipped with a 325 nm narrow band filter, a 2 nm full-width at half-maximum, and a transmission that was greater than 75% at 325 nm. Under the excitation of 325-nm, PL spectra and the ‘kinetics’ mode of the UV emission intensity were measured using a spectrofluorometer (HORIBA, Fluoromax-4) equipped with a 2L vacuum chamber.

A temperature controller system was used to precisely tune the sample temperature. Oxygen and nitrogen were introduced into an air chamber through an air mixing system. A mechanical pump was used to discharge gas into the air chamber. On the one hand, the optical gas-sensing test system can collect the typical PL spectra of samples with different oxygen content. On the other hand, it can also use the “dynamics” mode of fluorescence spectrophotometer to provide real-time information of PL-based gas sensing by monitoring the intensity of UV emission at any wavelength, and the time resolution of spectrum collection is 0.1s. The oxygen-sensing response of the samples at any temperature was defined as R = (*I*_oxygen_ − *I*_nitrogen_)/*I*_nitrogen_, where *I*_oxygen_ and *I*_nitrogen_ are the maximum values of the UV emission intensity measured in oxygen and air atmospheres, respectively.

### 2.3. Material Models and DFT Simulation

We selected O_2_ molecules as the adsorption models and the (002) surface of ZnO and the (002) surface of Au/ZnO as the adsorption interface to study the adsorption mechanism. In the Au/ZnO model, the most stable configuration was that of Au on the surface of ZnO nanorods. A ZnO supercell (4 × 4 × 2) was used. Different adsorption sites and configurations of Au (111) and Au (200) were examined; the most stable Au/ZnO surface was derived from an orientation of Au that was approximately parallel to the surface.

We used density functional theory (DFT) to conduct the first principal calculation using VASP (Vienna Ab-initio Simulation Package) software. In this study, the interaction between the valence electrons and ion nuclei was projected as an extended wave. Perdew–Burk–Ernzerhof (PBE) under the general gradient approximation (GGA) was used to manage the exchange correlation energy between electrons. The plane-wave truncation energy was set to 400 eV, and point K in the Brillouin zone was set to 3 × 3 × 1. To avoid interaction between the layers, we set a vacuum layer of 15 Å. The van der Waals interaction in the calculation was approximated using the DFT–D_3_ method. When the structure is optimized, the lattice parameters and atomic positions of the structure are relaxed; the energy convergence accuracy is 1 × 10^−5^ eV, and the convergence precision of the force applied to each atom is 0.01 eV/Å. The adsorption energy is expressed as
(1)$$E_{\mathrm{ads}}=E_{\mathrm{total}}-E_{\mathrm{slab}}-E_{o_{2}}$$
where $E_{\mathrm{total}}$ is the total energy of the system; $E_{\mathrm{slab}}$ is the base energy, and $E_{o_{2}}$ is the energy of adsorbed oxygen. The differential charge density is expressed as
(2)$$\rho_{\mathrm{diff}}=\rho_{\mathrm{total}}-\rho_{\mathrm{slab}}-\rho_{o_{2}}$$
where $\rho_{\mathrm{total}}$ is the total charge density of the system; $\rho_{\mathrm{slab}}$ is the base charge density, and $\rho_{o_{2}}$ is the charge density of adsorbed oxygen.

## 3. Results and Discussion

### 3.1. Morphology and Structure of Au/ZnO NR Samples

The morphologies and structures of the Au/ZnO NR samples synthesized at different Au sputtering times were characterized by SEM. Figure 2a–c presents top-view SEM images of Au/ZnO NR with Au sputtering times of 0 s, 50 s, and 90 s, respectively. The ZnO NR array samples exhibited an obvious hexagonal prism morphology with an average diameter (*d*) of ~100 nm and excellent *c*-axis-preferred orientation. When the Au sputtering time was increased to 90 s, there were more Au NPs on the top and upper parts of the ZnO NR array. As shown in Figure 2d–f, the ZnO NR arrays grew perpendicular to the Si substrate and had an excellent *c*-axis-preferred orientation. The height of the ZnO NR array was approximately 1 µm. In contrast, an increasing number of Au NPs gathered on the top and profile of ZnO NR with increased Au sputtering time. As shown in Figure 2f, a large number of Au particles were stacked on top of the ZnO NR sample, which exhibited a hemispherical shape. After considering the Au sputtering time and morphology of the nanorods, ZnO NR array substrates were chosen for subsequent characterization experiments.

As shown in Figure 2g–i, the EDS spectra used to analyze the elemental composition of the Au/ZnO NR array substrates confirmed that Zn, O, Au, and Si were the main elements in these samples, and the Zn:O ratio was close to 1:1. Significantly, the atomic content of oxygen atoms was lower than that of zinc atoms, which indicates that there were many oxygen defects in these samples. The defects in oxygen species were mainly caused by the hydrothermal growth of ZnO NR. In addition, as the Au sputtering time increased, the content of Au atoms gradually increased, indicating that increasing the sputtering time could cause the nanorods to load more Au NPs and regulate their distribution on the surface of ZnO NR array samples.

To further understand the morphology and crystal structure of the Au/ZnO nanorods, TEM measurements revealed that the Au/ZnO NR samples comprised a shell core structure of Au-coated ZnO nanorods with an average size of ~100 nm (Figure 3a). The crystal structures of the ZnO NR and Au/ZnO NR samples were further characterized using XRD. As shown in Figure 3b, the XRD patterns of the ZnO NR and Au/ZnO NR samples with a 50-s Au sputtering time show diffraction peaks that can be indexed to the (002) planes corresponding to the ZnO hexagonal phase crystal structure, with no diffraction peaks of any impurity phases detected. When the Au sputtering time reached 90 s, diffraction peaks of the Au (111) and (200) planes appeared in the XRD patterns. The TEM and XRD data confirmed that the synthesized ZnO NR samples possessed good crystalline quality.

### 3.2. Optical Properties of Au/ZnO NR Samples

The PL spectra of the ZnO NR arrays and Au/ZnO NR array samples measured in air at room temperature are shown in Figure 4 and Figure 5. The intrinsic optical characteristics of the ZnO NR samples were analyzed by PL measurements to investigate the native structural defects of the ZnO NR and Au/ZnO NR nanostructures. The PL spectra peaks centered at ~390 nm in the UV emission region originate from the intrinsic band-to-band transition emission, and the visible-band emission of the ZnO NR sample is dominated by an oxygen-deficiency-related component [22,23]. The ZnO NR samples prepared by the hydrothermal method contained more hydroxyl, oxygen vacancies, oxygen enrichment, and other defects in the interior and on the surface, which increased the peak intensity of visible components in the PL spectrum [22,23]. The main objective of this work was to use the ultraviolet component in the PL spectrum of the ZnO NR samples as the monitoring object for gas-sensing measurements. The crystal quality of the gas-sensing material must be improved, with fewer structure-related defects for an excellent response. Thus, the original ZnO NR array prepared using the hydrothermal method was annealed. As shown in Figure 4 when ZnO NR samples with different annealing temperatures (300 °C, 500 °C, 700 °C in Ar atmosphere) were excited at 325 nm, the ratio of the peak intensities of the UV emission and visible band (*I*_UV_/*I*_visible_) from the high-temperature annealed ZnO sample was much larger than that of the unannealed ZnO sample, implying that the annealed ZnO sample had a higher crystalline quality, and thus, a higher UV emission efficiency luminescence peak intensity. It is worth noting that the ~570 nm peak at the unannealed PL spectrum will cause a blue shift to ~500 nm when the annealing temperature increases. This can be attributed to the oxygen-rich related bulk defects and surface-related defects (~570 nm peak) decreasing, while the defects related to oxygen vacancy (~500 nm peak) increase when the ZnO samples are annealed at a high temperature in Ar atmosphere.

To explore the effect of the PL characteristics of Au NPs, ZnO NR arrays were sputtered with Au for different times (0–90 s). As shown in Figure 5, the intensity of the UV emission peak first increases and then decreases with increasing Au sputtering time. The ZnO NR array sample annealed at 700 °C had the best luminous characteristics when the NR surfaces were modified Au NPs with a sputtering time of approximately 50 s (maximum UV peak intensity and maximum *I*_UV_/*I*_visible_). It can be understood that the defect-state electron concentration of the sample annealed at 700 °C is much lower than that of samples annealed in other conditions. If we continue to increase the sputtering time of Au NPs, the luminescence characteristics of the UV emission decrease, mainly because the exciton concentration can be increased by transferring the electron conduction band of the defective energy level through the surface plasma effect to improve the luminescence characteristics when the ZnO sample surfaces are modified by Au NPs. [46] In addition, a part of the ZnO NR surface is covered by Au NPs, which reduces the area of stimulated luminescence. The surface-attached Au NPs also reflected and absorbed the luminescence of the samples, which decreased their luminescence characteristics. These two mechanisms affect the luminescence characteristics of Au/ZnO NR samples.

### 3.3. Gas-Sensing Properties of Au/ZnO NR Sample

Temperature (*T*) affects the PL intensity of the Au/ZnO NRs and their gas-sensing characteristics. When the temperature rises, the target gas receives more energy; the surface defect activity of the sample is improved, and the concentration of the gas adsorption site is increased, which contributes to improving the reaction sensitivity of the sample to detect gas and effectively shortens its adsorption and desorption times [47,48,49]. PL spectra of the Au/ZnO NR samples were measured in a 20% oxygen atmosphere at different temperatures (*T*). Figure 6a shows the *T*-dependence of response (R) for Au/ZnO NR samples from *T* = 100 °C to *T* = 300 °C. The R increases up to ~225 °C, but decreases thereafter; the largest R value is approximately 36%. This behavior can be ascribed to the adsorption of water-related species and surface reactions. Considering *I*_UV_ and R, 225 °C was selected as the optimal working temperature for subsequent gas-sensing experiments. The oxygen-sensing performance of the Au/ZnO NR samples was evaluated by measuring their PL in pure nitrogen, pure oxygen, and in atmospheres with different contents of nitrogen and oxygen. The PL measurements in Figure 6b at optimal working temperatures show that the *I*_UV_ emission peak intensity of the Au/ZnO NR samples significantly decreased as pure oxygen was added to the detection cavity. The R value reached approximately 50% in a 100% oxygen atmosphere.

The dynamic *I*_UV_ (~390 nm) response of the Au/ZnO NR samples to oxygen at *T* = 225 °C was investigated by varying the oxygen content mixed with nitrogen from 10% to 100%. As shown in Figure 6c, *I*_UV_ decreases with increasing oxygen content and yields an easily measurable response at a low oxygen gas content of 10%. Figure 6d shows that the response of the oxygen content in the 10–100% range has a good linear relationship of *y* = 0.172*x* + 29.473, which can be reasonably used to measure a wide range of oxygen concentrations under normal pressure.

In order to evaluate the sensitivity, detection limit, and response time of the Au/ZnO NR sample, the dynamic *I*_390_ response of the Au/ZnO NR sample to oxygen at *T* = 225 °C and *P* = 1 bar was investigated by changing the oxygen gas content in nitrogen from 1 to 5%. As Figure 7 shows, *I*_390_ decreases with an oxygen gas content increase and yields a clearly measurable response even at a low oxygen gas content of 1%. The response values R of 1%, 2%, and 5% oxygen content are 5.8, 10.7, and 14.2, respectively. The measured response and recovery times of the Au/ZnO NR sample are 2 and 3 min with gas concentrations of 2% oxygen content, respectively. These experimental data confirm the good stability, sensitivity, recoverability, and reproducibility of oxygen sensing, and illustrate the potential of Au/ZnO NR samples as an optical oxygen sensor.

In addition, selectivity is very important for gas sensors in practical applications. The responses of 10% H_2_O, CH_4_, NO_2_, CO_2_, H_2_, NH_3_, and O_2_ gasses were investigated by the same test method, as shown in Figure 8a. Except for O_2_ gas (R = 31.1%), the responses of these gases were less than 5%, indicating that this O_2_ gas sensor has very good selectivity. The long-term stability of the Au/ZnO NRs sensor towards 10%, 20%, and 40% oxygen in nitrogen gas was studied at an optimal working temperature. As shown in Figure 8b, the response value of the Au/ZnO NRs sensor has no distinct changes in 30 days, revealing excellent stability. These data further confirm the good reproducibility and selectivity of the O_2_ gas sensing and illustrate the promise of the Au/ZnO NRs samples as an optical oxygen sensor.

In order to visually demonstrate the high response of our O_2_ sensor, in Table 1 we summarize the oxygen-sensing performance of different sensitive materials. Different oxygen-sensing methods and the obtained sensing performance are compared in detail. Optical gas sensing is the most effective and widely used method to detect oxygen at present. Nanostructures of ZnO possess a much wider detection range, a low detection limit, good long-term stability, and better gas-sensing performance. Therefore, we will further improve the performance of gas sensors through the doped and surface modification of ZnO NRs sample.

### 3.4. Gas-Sensing Mechanism of Au/ZnO Sensors

It can be seen from Figure 9 that the modification of Au NPs on the surface of ZnO NRs results in the formation of a typical metal semiconductor Schottky barrier between Au and ZnO. Since ZnO (4.4 eV) is significantly smaller than the work function of Au (5.1 eV), more electrons are transferred from ZnO NRs to Au NPs until the Fermi level reaches equilibrium [50,51,52]. In Figure 9a,b, the Au/ZnO NR sensor diagram illustrates a mechanism for the outstanding gas-sensing capability of Au/ZnO NR samples. Several groups have suggested that the formation of adsorbed oxygen species such as O^2−^, O^−^, and O_2_^−^ leads to an electron depletion layer at the Au NPs, ZnO NR surface, and Au/ZnO NR interface [22,53,54]. Thus, the thickness of the electron depletion layer increases with increasing oxygen content. O^−^ and O^2−^ ions are the dominant adsorbed oxygen species on the material surface [50,51,52,53,54]; oxygen sensing by the Au/ZnO NR sample is illustrated in Figure 6b. Based on the research, the reaction mechanism can be depicted by the following equations:(3)$$O_{2}\;(\mathrm{gas})\;\to O_{2}\;\left(\mathrm{ads}\right)$$
(4)$$O_{2}\;\left(\mathrm{ads}\right)+e^{-}\to O_{2}{}^{-}\;\left(\mathrm{ads}\right)$$
(5)$$O_{2}{}^{-}\;(\mathrm{ads})+e^{-}\to 2O_{2}{}^{-}\;\left(\mathrm{ads}\right)$$
(6)$$O^{-}\;(\mathrm{ads})+e^{-}\to O^{2}{}^{-}\;\left(\mathrm{ads}\right)\;$$

According to the literature, the expansion or contraction of the electron depletion layer on the surface of ZnO is attributed to the adsorption/desorption of oxygen species in the environment, which can capture or release electrons [47,48,49,50]. Since the depletion layer has a great influence on the visible emission of ZnO and has little contribution to the UV emission, the UV emission intensity mainly depends on the volume of the “non depletion” area below the depletion layer. Therefore, the response of the oxygen sensor based on *I*_390_ mainly depends on the surface and interface conditions of ZnO-based nanomaterials. Based on this sensing mechanism, possible reasons for the superior oxygen-sensing performance of Au/ZnO NR samples include: (1) Au/ZnO NR samples have a much larger surface area to volume (*S*/*V*) ratio (Figure 2); (2) the Au NPs and Au/ZnO NR interface introduce additional active sites on the sample surface that facilitate oxygen adsorption [49,50,51,52]; (3) under 325-nm UV light excitation, Au NPs provide an effective channel for electron transfer on the interface of Au/ZnO NRs and accelerate the reaction when the sample is exposed to an oxygen atmosphere. In addition, the appropriate coordination structure of Au NPs on the ZnO surface for oxygen gas molecules is a vital factor for enhancing the response and selectivity of composite sensors to oxygen [50,51,52]. The high-temperature annealed PL data indicate that the Au/ZnO NR samples have a higher crystalline quality, and thus, a higher UV emission efficiency. High UV emission efficiency and emission intensity are crucial for high-performance optical-based gas sensing because sensing depends on accurate measurement of emission intensity variation. Controlling the content and morphology of Au NPs can help improve the optical oxygen-sensing properties of ZnO NR samples.

To further confirm the gas-sensing mechanism, DFT calculations were performed using Material Studio software. The analysis and calculation of the model geometry and energy parameters theoretically verified the increase in the sensitivity of oxygen molecules after Au NP modification of ZnO NRs. In DFT simulations, the ZnO (002) surface is often studied as one of the most stable surfaces [55,56,57]. The optimized configurations before and after O_2_ molecule adsorption on the Au/ZnO surface are shown in Figure 10a,b. Figure 10b shows the optimized Au/ZnO configurations after adsorbing oxygen molecules. Green and yellow indicate the effective negative and positive charges, respectively. As shown in Figure 10b, the charge transfer occurs in the contact area. The increase in the green contact area near adsorbed oxygen molecules indicates that many electrons are transferred from ZnO and Au to oxygen molecules, which indicates that there is a strong interaction between them [56,57,58]. DFT simulation results showed that the atomic position at the surface and interface significantly changed after the adsorption of oxygen molecules, indicating that Au NPs successfully modified ZnO and changed the charge density on the material surface and interface.

The optimized configurations of O_2_ molecule adsorption on the Au (200), Au (111), ZnO (002), and Au/ZnO surfaces are shown in Figure 10 and Figure 11. The adsorption energies of the optimal configurations are presented in Table 2. It is observed that oxygen can be adsorbed on clean ZnO, Au, and Au/ZnO surfaces. ZnO, Au, and Au/ZnO materials easily adsorb oxygen; ZnO has the best adsorption effect on oxygen. It is known that a larger surface area implies a greater adsorption capacity, originating from high-density ZnO NR arrays with excellent *c*-axis orientation. In reality, the adsorption surface area decreases with increasing amounts of adsorbate. Thus, the Au/ZnO composite structure can easily adsorb oxygen molecules.

To conduct an in-depth and systematic study of the electronic structure characteristics of O_2_ molecules before and after adsorption on clean ZnO, Au, and Au/ZnO surfaces, the total state density (TDOS) and partial state density (PDOS) plots of the ZnO, Au, and Au/ZnO adsorption systems were investigated. The TDOS and PDOS images before and after adsorption of O_2_ molecules in the optimal configurations are shown in Figure 12a–d. Compared with the ZnO adsorption system, the TDOS diagram shows that the electron density of the Au/ZnO adsorption system is significantly enhanced at −4.5 eV, −4.17 eV, −3.0 eV, −1.5 eV, −1.27 eV, −0.36 eV, and 0.08 eV, mainly due to the influence of the Au-5d and Zn-3d orbitals (Figure 12a,b). As shown in Figure 12c,d, the TDOS of Au/ZnO significantly decreased near −4.5 eV and −3 eV after O_2_ molecule adsorption due to strong hybridization between oxygen and Au-ZnO. Interaction of the Au-5d, Zn-3d, and O-2p orbitals occurred via overlap near −4.5 eV, −4.17 eV, −3.0 eV, −1.5 eV, −1.27 eV, −0.36 eV, and 0.08 eV in PDOS. These results show that surface modification of Au NPs and adsorption of oxygen on the Au/ZnO surface can transfer the charge at the surface and interface, resulting in good gas-sensing properties. These DFT simulation results are consistent with our gas sensitivity test results.

## 4. Conclusions and Future Work

Novel optical gas-sensing materials for Au NP-modified ZnO NR arrays were fabricated by hydrothermal synthesis and magnetron sputtering. Such gas sensors have excellent optical oxygen-sensing properties at 225 °C, with a demonstrated response reaching ~55% and a wide linear detection range of oxygen content from 10% to 100%. The proposed oxygen-sensing strategy can be used in the promotion of life sciences and environmental quality, as well as automotive, chemical, and aerospace technology. Through DFT simulations, we verified that oxygen was adsorbed on the Au/ZnO surface and interface, which was the main cause of enhanced sensing properties. This study provides a demonstration of a dynamic PL-based oxygen gas sensor and suggests a method using PL spectroscopy to study the optical properties of Au NPs-modified ZnO NRs. This work provides a highly sensitive, very promising, low-cost approach for gas sensing. At the same time, this work also lays a foundation for the development of metal-sensitized semiconductor oxides in the research and development of gas sensor electronic devices.

Over time, the role of gas sensors has become more and more important. Experts from all fields, from engineering to materials science, are studying metal oxide gas sensors. In the future, the design and fabrication of a flexible substrate, complex gas environment detection, remote detection, and multi-target integrated gas sensor chips will become an increasingly key research direction for gas sensors. Room temperature, remote, fast response, multi-target real-time monitoring, and special test environments are still major challenges for gas sensors. Optical gas sensors have unique advantages in the field of remote detection. The construction of n-n, n-p, and metal-sensitized semiconductor heterostructures and complex multi-dimensional structure-sensing composites is crucial to solve the problem of highly sensitive gas monitoring under special environmental conditions. It is an important means to improve the sensitivity and selectivity of gas sensors to modify semiconductor oxides by precisely controlling the morphology and structure of noble metals. The combined geometry of Au, ZnO has offered more active sites for O_2_ adsorption, attributed to the formation of a Schottky contact between the Au–ZnO. The Au can act as an electron sink, which extracts electrons from the conduction band of the ZnO and facilitates the electron transfer. Therefore, designing and constructing reasonable band structure models and surface models of gas sensors is one of the most effective ways to improve gas-sensing performance.

## Figures and Tables

**Figure 1 sensors-23-02886-f001:**
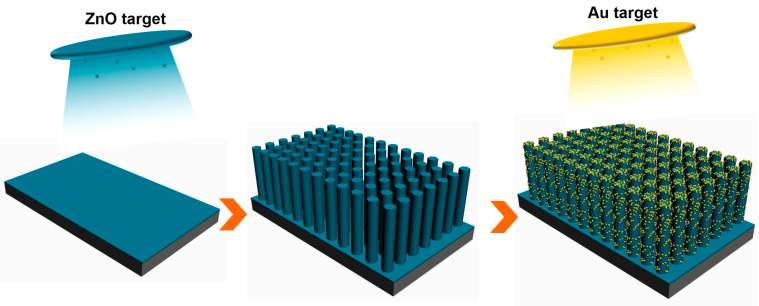
Preparation process of Au/ZnO NRs array substrate.

**Figure 2 sensors-23-02886-f002:**
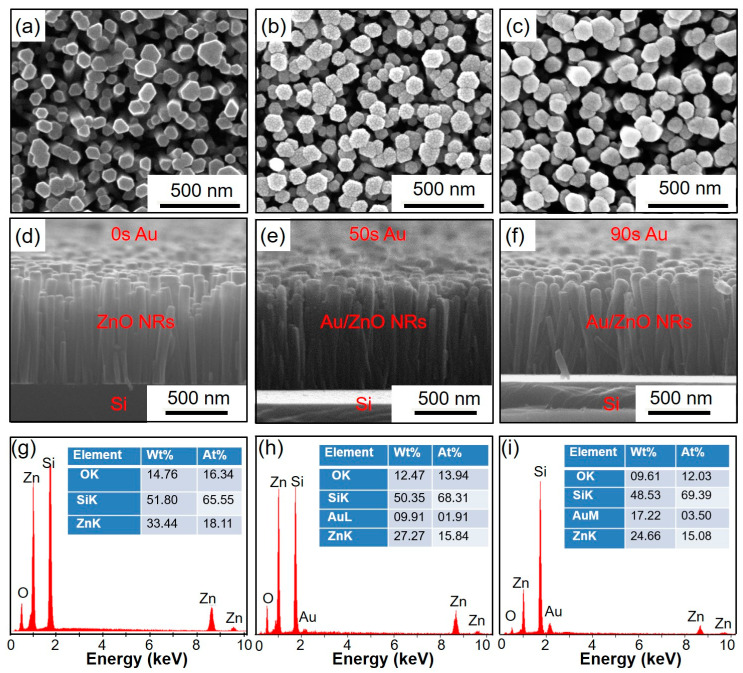
(**a**–**f**) Top and profile view SEM images of ZnO NR with different Au sputtering times. (**g**–**i**) EDS spectra of ZnO NR with different Au sputtering times.

**Figure 3 sensors-23-02886-f003:**
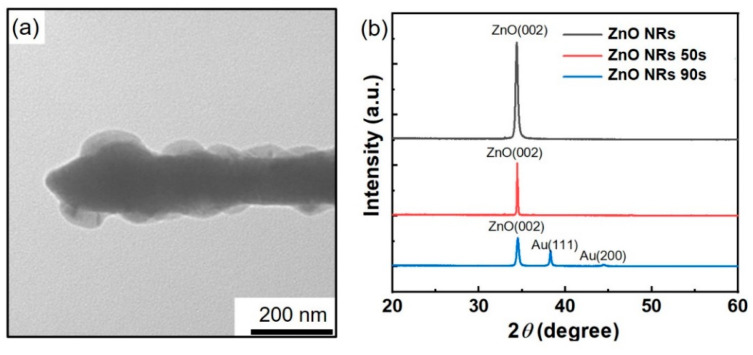
(**a**) TEM image of Au nanoparticles modified with single ZnO nanorod and (**b**) XRD patterns of ZnO NRs with different Au sputtering times.

**Figure 4 sensors-23-02886-f004:**
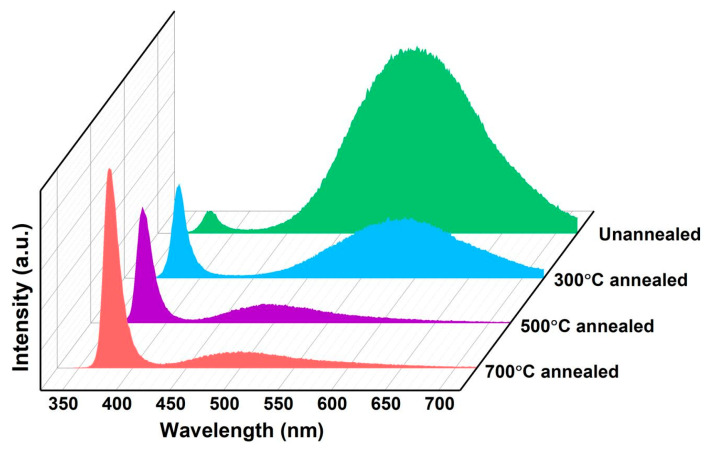
PL spectra of ZnO NRs at different annealing temperatures.

**Figure 5 sensors-23-02886-f005:**
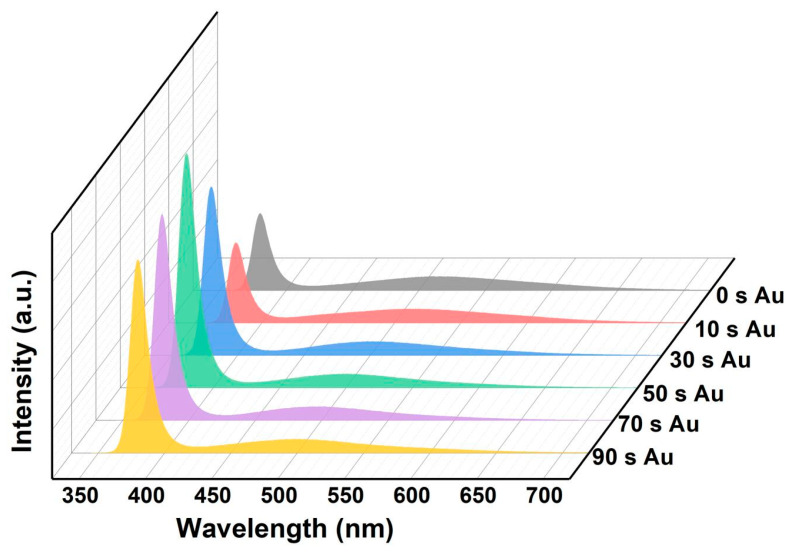
PL spectra of ZnO NRs with different Au sputtering times.

**Figure 6 sensors-23-02886-f006:**
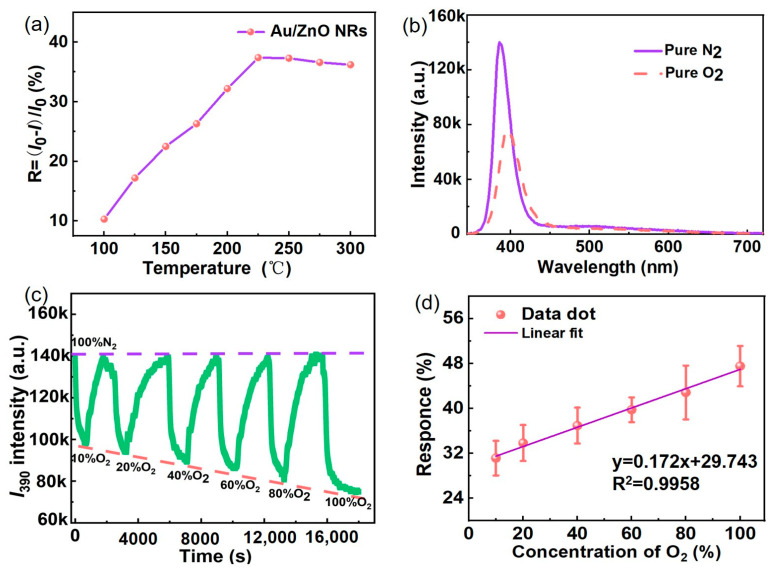
(**a**) Responses of Au/ZnO NRs with exposure to 20% oxygen atmosphere at temperatures ranging from 100–300 °C. (**b**) PL spectra of Au/ZnO NRs measured in pure nitrogen and oxygen atmospheres. (**c**) Dynamic response of Au/ZnO NRs with exposure to 10–100% oxygen atmospheres at 225 °C. (**d**) Fitting line of responses of Au/ZnO NRs with exposure to 10–100% oxygen atmospheres.

**Figure 7 sensors-23-02886-f007:**
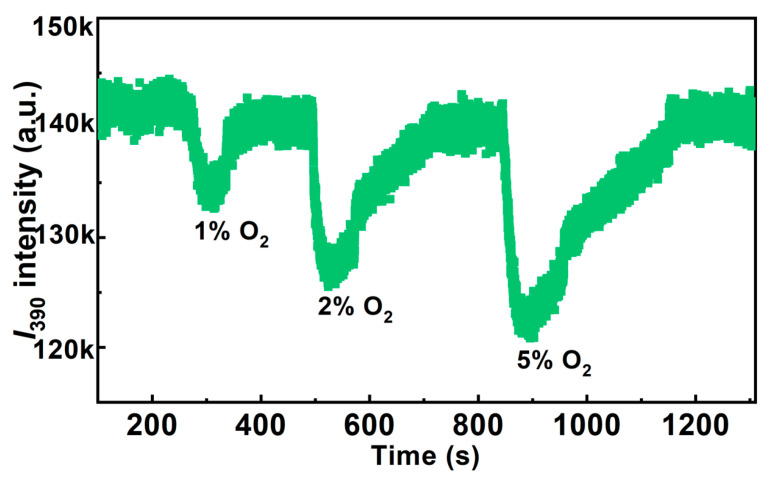
Dynamic response of Au/ZnO NRs with exposure to 1–5% oxygen atmospheres.

**Figure 8 sensors-23-02886-f008:**
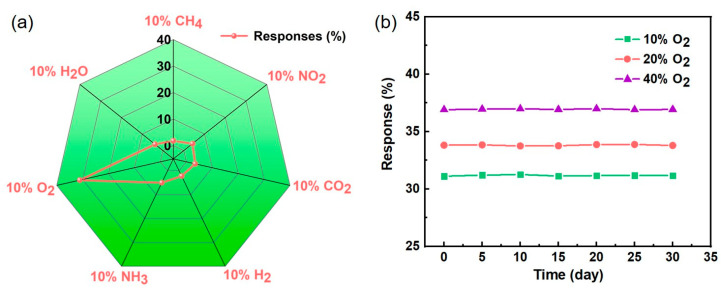
(**a**) Selectivity of Au/ZnO NRs sample upon exposure to different gasses in nitrogen. (**b**) Long-term stability of Au/ZnO NRs sensor upon exposure to 10%, 20%, and 40% oxygen in nitrogen gas.

**Figure 9 sensors-23-02886-f009:**
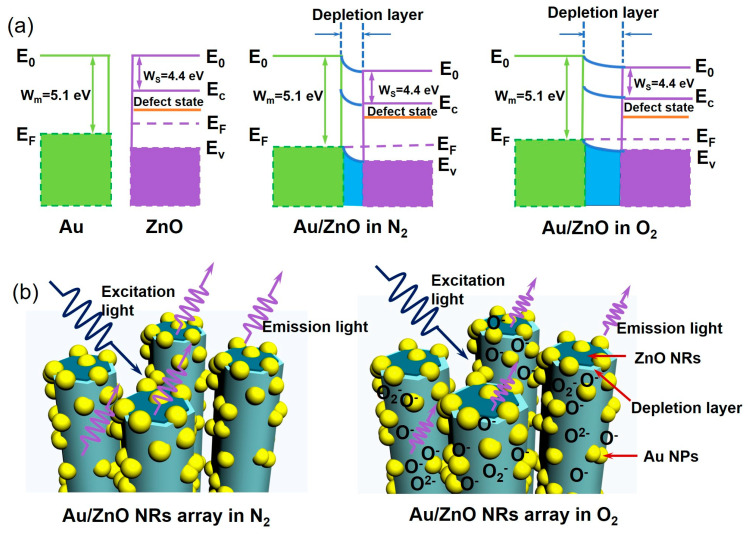
(**a**) Energy band and (**b**) fluorescence detection illustration of oxygen-sensing mechanism of Au/ZnO NRs.

**Figure 10 sensors-23-02886-f010:**
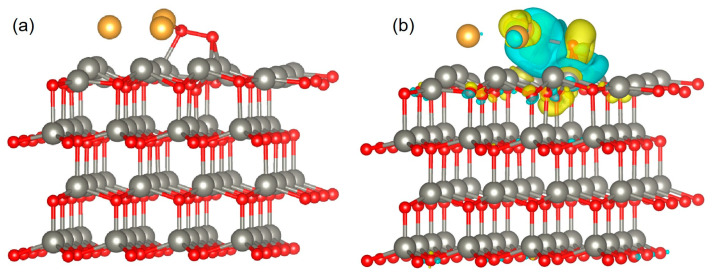
(**a**) Optimized configurations before O_2_ molecule adsorption on Au/ZnO surface. (**b**) Optimized configurations after O_2_ molecule adsorption on Au/ZnO surface and surface charge density change.

**Figure 11 sensors-23-02886-f011:**
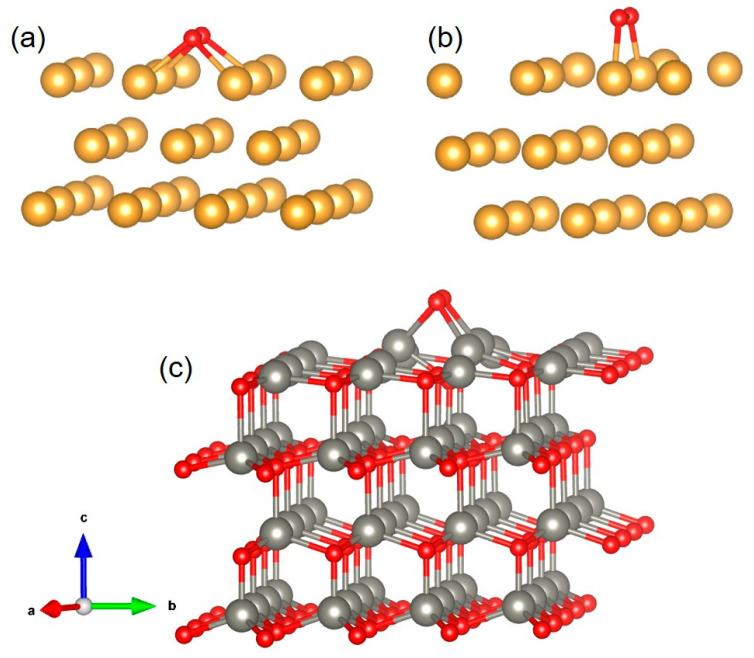
Optimized configurations of O_2_ molecule adsorption on (**a**) Au (200), (**b**) Au (111), and (**c**) ZnO (002) surface.

**Figure 12 sensors-23-02886-f012:**
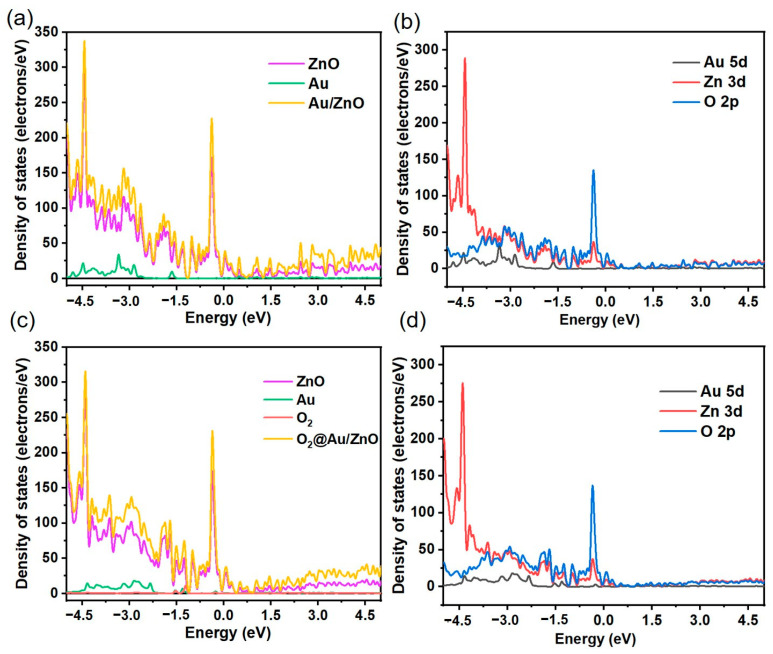
(**a**) TDOS and (**b**) PDOS before minor O_2_ molecule adsorption of ZnO, Au, and Au/ZnO. (**c**) TDOS and (**d**) PDOS after minor O_2_ molecule adsorption of ZnO, Au, and Au/ZnO.

**Table 1 sensors-23-02886-t001:** Summary of the Sensing Properties of the O_2_ Sensor.

Sensitive Materials	Method	Detection Range	Detection Limit	Stability	Reference
PAM-CS DN	Electrochemical	1–100% O_2_	1% O_2_	Temperature and humidity	[6], 2022
ZnO/CdS-EDTA	Electrical	10–50 ppm	10 ppm	Long-term	[8], 2016
ZnO nanowires	Photoluminescence	9% O_2_ (10–50 sccm)	9% O_2_	Flow	[10], 2016
Dye	Optochemical	5–100% O_2_	5% O_2_		[11], 2008
PVDC	Photoluminescence	2–30% O_2_	2% O_2_	Temperature	[14], 2019
Pr^3+^/(K_0.5_ Na_0.5_)Nb_3_	Photoluminescence	2–100% O_2_	2% O_2_		[18], 2016
Pd/TiO_2_	Electrical	100–1000 ppm	100 ppm	Flow	[20], 2007
Mesoporous TiO_2_	Photoluminescence	2–20% O_2_	2% O_2_		[21], 2021
Pr/ZnO nanofibers	Photoluminescence	3–20% O_2_	3% O_2_		[22], 2019
Au/ZnO nanorods	Photoluminescence	1–100% O_2_	1% O_2_	Long-term	This work

**Table 2 sensors-23-02886-t002:** The calculated adsorption energies of O_2_ on various surfaces.

Structure	*E* _total_	*E* _O2_	*E* _base_	*E* _adsorption_
Au (200)	−103	−9.85	−92.209	−0.941
Au (111)	−107	−9.85	−96.191	−0.959
ZnO (002)	−574	−9.85	−559.87	−4.28
Au/ZnO	−585	−9.85	−573.62	−1.53

## Data Availability

Not applicable.

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
