# Peer review of "Novel Au Nanoparticle-Modified ZnO Nanorod Arrays for Enhanced Photoluminescence-Based Optical Sensing of Oxygen"

_sensors, 2023, doi:10.3390/s23062886_

Round 1
Reviewer 1 Report
The manuscript entitled “Novel Au Nanoparticle-modified ZnO Nanorod Arrays for Enhanced Photoluminescence-based Optical Sensing of Oxygen” by Baosheng Du et al is well documented and written. The materials are well characterized and the experiments were performed according to proper scientific methods, but there are some questions.
· • Although there are some experiments concerning the selectivity in comparison with the CH4, NO2, CO2, H2, NH3, measurements also for the H2O are needed. Moreover, it would be very useful to measure the O2 in the presence of some of these gases.
· Some measurements of short and/or long-term stability of the sensor could be very useful.
· Please give the intensity of the excitation radiation and/or the source of the radiation.
• At line 174 please check the “and the Zn:P ratio was close to 1:1.”
• At lines 199-200 “The PL spectra of the ZnO NR arrays and Au/ZnO NR array samples measured in air 199 at room temperature are shown in Figures 4a and b, respectively” but there is only a Figure 4 no a or b please correct it.
For these reasons I propose that the manuscript be accepted for publication after answering to the previous questions/suggestions.
Author Response
Dear Reviewer,
Re: Manuscript ID: 2229239
Thank you for the reviewer’s comments concerning our manuscript entitled “Novel Au Nanoparticle-modified ZnO Nanorod Arrays for Enhanced Photoluminescence-based Optical Sensing of Oxygen”. Those comments are all valuable and very helpful for revising and improving our paper. We have studied these comments carefully and have made correction according the comments, which we hope meet your approval. Revised portion are marked in red in the manuscript. The main corrections in the paper and the responds to the reviewer’s comments are as flowing:
Reviewer #1:
Comments No. 1:
- Although there are some experiments concerning the selectivity in comparison with the CH4, NO2, CO2, H2, NH3, measurements also for the H2O are needed. Moreover, it would be very useful to measure the O2 in the presence of some of these gases.
- Some measurements of short and/or long-term stability of the sensor could be very useful.
- Please give the intensity of the excitation radiation and/or the source of the radiation.
Response:
Thanks to Reviewer for reminder, we have added the H2O to discuss the selectivity of oxygen detection at line 281 in the manuscript. The long-term stability of the sensor was analyzed in Figure 8b and discussed at line 286-290 in the manuscript. The radiation source is 150 W ozone free Xe lamp equipped with 325 nm narrow band filter, a 2 nm full-width at half-maximum, and transmission that is greater than 75% at 325 nm.
Comments No. 2:
- At line 174 please check the “and the Zn:P ratio was close to 1:1.”
Response:
We are very sorry for our incorrect writing, we have revised the “Zn:P” to “Zn:O” in the manuscript.
Comments No. 3:
- At lines 199-200 “The PL spectra of the ZnO NR arrays and Au/ZnO NR array samples measured in air 199 at room temperature are shown in Figures 4a and b, respectively” but there is only a Figure 4 no a or b please correct it.
Response: We are very sorry for our incorrect writing, we have revised the “Figures 4a and b” to “Figure 4 and Figure 5” in the manuscript.
Special thanks to you good comments.

Reviewer 2 Report
This work introduced a gold nanoparticle modified ZnO nanorod detector for oxygen. This paper is well written and organized, may be accepted after minor revisions solving questions as follow.
1. Figure 4. It seems that UV peak intensity of PL spectrum keeps increase when we increase the annealing temperature from 300K to 973K. Would authors please answer that will this tendency keep, or the UV peak intensity will decrease after reaching a critical annealing temperature?
2. Figure 4. It seems that the 550nm peak at unannealed spectrum will make blue shift to 500nm while annealing temperature goes high. Can authors explain this?
3. If authors can make more comparison between their method and other methods will be largely appreciated. Like the precision, price, stability.
Author Response
Dear Reviewer,
Re: Manuscript ID: 2229239
Thank you for the reviewer’s comments concerning our manuscript entitled “Novel Au Nanoparticle-modified ZnO Nanorod Arrays for Enhanced Photoluminescence-based Optical Sensing of Oxygen”. Those comments are all valuable and very helpful for revising and improving our paper. We have studied these comments carefully and have made correction according the comments, which we hope meet your approval. Revised portion are marked in red in the manuscript. The main corrections in the paper and the responds to the reviewer’s comments are as flowing:
Comments No. 1:
- Figure 4. It seems that UV peak intensity of PL spectrum keeps increase when we increase the annealing temperature from 300K to 973K. Would authors please answer that will this tendency keep, or the UV peak intensity will decrease after reaching a critical annealing temperature?
Response: Your question is very good. Since we did not express it clearly. We are sorry for your misunderstanding. the UV peak intensity will decrease after reaching about 700°C annealing temperature. In fact, the high temperature fluorescence characteristic of ZnO comes from its large exciton binding energy, which makes it more stable at higher temperature or under the action of electric field. When the temperature is high enough(>700°C), the exciton binding ability is destroyed and the UV emission peak intensity will decrease. Relevant references are as follows: Yongqi Yin, Ye Sun, Miao Yu, Xiao Liu, Bin Yang, Danqing Liu, Shaoqin Liu, Wenwu Cao, Michael N. R. Ashfold*, Arrays of nanorods composed of ZnO nanodots exhibiting enhanced UV emission and stability, Nanoscale, 2014, 6(18): 10746-10751.
Comments No. 2:
- Figure 4. It seems that the 550nm peak at unannealed spectrum will make blue shift to 500nm while annealing temperature goes high. Can authors explain this?
Response: Your question is very good. Since we did not express it clearly. We are sorry for your misunderstanding. Emission centred at ~500 nm of ZnO samples is usually assigned to oxygen vacancy defects, while the longer wavelength emission (550–610 nm range) is traditionally associated with defects based on oxygen-rich related bulk defects (e.g. oxygen interstitials) and/or to surface-related defects (e.g. adsorbed oxygen or water molecules). When the ZnO samples is annealed at high temperature in Ar atmosphere, the oxygen-rich related bulk defects and surface-related defects will decrease, while the defects related to oxygen vacancy will increase. Therefore, the 550nm peak at unannealed spectrum will make blue shift to 500nm while annealing temperature goes high. The corresponding discussion has added in the manuscript at line 215-219. Relevant references are as follows:
- Yongqi Yin, Ye Sun, Miao Yu, Xiao Liu, Bin Yang, Danqing Liu, Shaoqin Liu, Wenwu Cao, Michael N. R. Ashfold*, Arrays of nanorods composed of ZnO nanodots exhibiting enhanced UV emission and stability, Nanoscale, 2014, 6(18): 10746-10751.
- Y. Yin, Y. Sun, M. Yu, X. Liu, B. Yang, D. Liu, S. Liu, W. Caoand M. N. R. Ashfold, Reagent concentration dependent variations in the stability and photoluminescence of silica-coated ZnO nanorods, Inorg. Chem. Front., 2015, 2(1), 28-34.
Comments No. 3:
- If authors can make more comparison between their method and other methods will be largely appreciated. Like the precision, price, stability.
Response: Thanks for your valuable suggestion, the comparison of detection method and O2 sensing performance has been added to Table 1 of the manuscript, and the relevant description is in line 297-304.
Special thanks to you good comments.

Round 2
Reviewer 1 Report
Please add the following information in the text "The radiation source is 150 W ozone free Xe lamp equipped with 325 nm narrow band filter, a 2 nm full-width at half-maximum, and transmission that is greater than 75% at 325 nm." In case that somebody wants to repeat the photoluminescence experiments.
Author Response
Dear Reviewer,
Re: Manuscript ID: 2229239
Thank you for the reviewer’s comments and suggestions concerning our manuscript entitled “Novel Au Nanoparticle-modified ZnO Nanorod Arrays for Enhanced Photoluminescence-based Optical Sensing of Oxygen”. Those comments are all valuable and very helpful for revising and improving our paper. We have studied these comments and suggestions carefully and have made correction according the comments, which we hope meet your approval. Revised portion are marked in red in the manuscript. The corrections in the paper and the responds to the reviewer’s comments are as flowing:
Reviewer #1:
Comments No. 1:
- Please add the following information in the text "The radiation source is 150 W ozone free Xe lamp equipped with 325 nm narrow band filter, a 2 nm full-width at half-maximum, and transmission that is greater than 75% at 325 nm." In case that somebody wants to repeat the photoluminescence experiments.
Response:
Thanks to Reviewer for reminder, we have added the information "The radiation source is 150 W ozone free Xe lamp equipped with 325 nm narrow band filter, a 2 nm full-width at half-maximum, and transmission that is greater than 75% at 325 nm"at line 117-119 in the manuscript.
